# Towards Trustful Machine Learning for Antimicrobial Therapy Using an Explainable Artificial Intelligence Dashboard

**Thomas De Corte** [1,2,†], **Jarne Verhaeghe** [3,†], **Femke Ongenae** [3], **Jan J. De Waele** [1,2] **and Sofie Van Hoecke** [3,*]

1 Department of Internal Medicine and Pediatrics, Faculty of Medicine and Health Sciences, Ghent University, 9052 Ghent, Belgium; thomas.decorte@ugent.be (T.D.C.)
2 Department of Intensive Care Medicine, Ghent University Hospital, 9000 Ghent, Belgium
3 IDLab, Ghent University-imec, 9052 Ghent, Belgium; jarne.verhaeghe@ugent.be (J.V.); femke.ongenae@ugent.be (F.O.)
* Correspondence: sofie.vanhoecke@ugent.be
† These authors contributed equally to this work.

**Abstract**

The application of machine learning (ML) in healthcare has surged, yet its adoption in high-stakes clinical domains, like the Intensive Care Unit (ICU), remains low. This gap is largely driven by a lack of clinician trust in AI decision support. Explainable AI (XAI) techniques aim to address this by explaining how an AI reaches its decisions, thereby improving transparency. However, rigorous evaluation of XAI methods in clinical settings is lacking. Therefore, we evaluated the perceived explainability of a dashboard incorporating three XAI methods for an ML model that predicts piperacillin plasma concentrations. The dashboard was evaluated by seven ICU clinicians using five distinct patient cases. We assessed the interpretation and perceived explainability of each XAI component through a targeted survey. The overall dashboard received a median score of seven out of ten for completeness of explainability, with Ceteris Paribus profiles identified as the most preferred XAI method. Our findings provide a practical framework for evaluating XAI in critical care, offering crucial insights into clinician preferences that can guide the future development and implementation of trustworthy AI in the ICU.

**Keywords:** machine learning; explainable artificial intelligence; dashboard; antimicrobial concentration; healthcare; intensive care unit

## 1. Introduction

An ever-increasing number of researchers and companies are developing applications to improve patient care based on artificial intelligence (AI) and machine learning (ML) [1]. One of the fields of interest is the intensive care unit (ICU), as it is a data-rich environment where the stakes for the individual patient are high. Despite great optimism about the potential of AI to provide substantial improvements in all areas of healthcare, only a few developed models have been adopted in clinical practice [2–4]. Besides legislative, organizational, and technical factors, the human factor is an important element to address when considering the implementation of innovative digital technologies [5,6]. Fostering end-user trust in AI outcomes is considered a crucial aspect of establishing clinical acceptance of AI and ML [7].

One of the proposed ways to increase trust is by using explainable AI models (XAI) [8]. These XAI methods provide insights into the decision-making processes of black-box

AI/ML models, promoting explainability—a necessary requirement in high-stakes environments [9]. Several XAI methods, such as LIME [10] or Shapley Additive explanations (SHAP-values) [11], have been proposed to provide (post hoc) explainability and clarity to the end-user [12]. Although several studies have evaluated the clinical effectiveness of medical AI, very limited research is available that specifically focuses on how these predictions should be communicated to clinicians through XAI [4,13–15]. To our knowledge, only one formal evaluation with clinicians exists that evaluated the use of XAI methods as a way to provide explainability [16]. Specifically, the study by Norrie et al. evaluated the usefulness of SHAP and LIME for clinicians in trusting sepsis prediction ML models [16]. In our study, we expand upon this study by adding a broader evaluation of different XAI methods and by providing more insight into the explainability requirements of AI for clinicians in the ICU.

To this end, we expand upon our previously developed Catboost regressor model [17]. The purpose of this model is to aid clinicians in optimizing the dosage of piperacillin, a commonly used antimicrobial in the ICU. We selected various XAI methods and integrated them into a dashboard to try to mimic real-world use in the ICU. The perceived explainability of applied XAI methods was evaluated by ICU clinicians using a questionnaire. The remainder of this article is structured as follows: Section 2 discusses the related work and the background required for the study. Section 3 then covers and elaborates upon different XAI methods, the use case, the used model, the software packages, the considered XAI methods, the dashboard, the questionnaire, and the recruited participants. The results of the study are discussed in Section 4, and finally, these results are then put into perspective in Section 5.

## 2. Background

A distinction is made between the interpretability and explainability of AI [18]. Interpretability tries to improve the understanding of the inner workings of ML algorithms, whereas explainability focuses on explaining the decisions made. The former answers the question of "how" an algorithm makes a prediction, while the latter provides information on "why" the prediction is made. Any technique helping to answer these questions generally falls under the term "explainable AI" (XAI).

There are numerous available XAI methods, which can be categorized in several ways [12,18,19]. A first classification is made based on the design of the AI/ML model itself and defines ad hoc and post hoc XAI methods or models. Ad hoc methods represent models that are intrinsically explainable by design (for example, representation or feature learning), whereas, for non-interpretable black or grey box models, one is limited to designing post hoc explanations. We will focus on the latter within this article. A second way of classifying XAI methods is according to the agnosticism of the model [12,18]. On the one hand, model-agnostic methods do not require access to the model architecture and are therefore applicable to all models [10,20]. They usually work by analyzing input and output pairs. Model-specific methods, on the other hand, are developed for a specific kind of ML model and consider the model's characteristics when determining explanations [21]. XAI techniques can also be categorized based on the scope of the explanation provided. Global methods attempt to explain the whole model, whereas local models focus on providing explanations for individual samples and predictions. Global methods will weigh input parameters the same way regardless of the individual prediction. Additionally, a distinction can be made between techniques that use complete instances to provide explanations [22–24] and techniques that focus on providing explanations using features [10,11,23]. Instance-based methods can be seen as row-based methods where they

use individual samples for explanations, in contrast to feature- or column-based methods that only look at features.

Determining whether or not an XAI method provides adequate explainability is challenging and mainly depends on the use case and the audience [19,25,26]. From a quantitative point of view, a conclusive test to determine the adequacy of XAI does not exist [18]. Instead, several parameters are used to evaluate explainability techniques [18]. Accuracy measures how well predictions made by the explainer model match the predictions of the complex model. Fidelity is determined by how correctly the explainer describes the behavior of the complex model. In addition to accuracy, explanations need to be truthful and complete to be considered fidel [19]. Stability refers to the robustness of the provided explanations to fluctuations when small changes occur on the input side [12]. The XAI method also needs to be deterministic: producing different explanations when applying the same method multiple times is undesirable. Finally, the explainer needs to be understandable for the end-user and cannot be too complex [19]. As the aforementioned parameters are a mix of quantitative and qualitative attributes, XAI evaluations are often performed by combining both research methods. While for the quantitative metrics, researchers fall back on the use of mathematical formulas most of the time, it is common practice to revert to standardized questionnaires with end-users for the objectification of qualitative metrics. Some of these human-centered evaluation methods are freely available online [27]. In this research paper, we focus on evaluating XAI understandability and, hence, will make use of qualitative research methods.

## 3. Materials and Methods

### 3.1. Use Case and Explainability Requirements

The use case in which the dashboard will be evaluated is the prediction of piperacillin plasma concentrations for infection management in critically ill patients in the ICU. Approximately 66% of patients admitted to the ICU receive antimicrobial therapy, with piperacillin being one of the most frequently prescribed antimicrobials [28,29]. Attainment of therapeutic antimicrobial plasma concentrations is believed to be beneficial for the patient by improving clinical outcomes and limiting drug toxicity. Target attainment is also believed to be beneficial for society by reducing the chance for antimicrobial resistance, which is considered one of the top 10 health *priori*ties by the World Health Organization [30]. Unfortunately, currently used dosing regimens are often insufficient to attain therapeutic plasma concentrations [31]. Adjusting the dosing regimen based on therapeutic drug monitoring (TDM), i.e., the measurement of the plasma concentration of the antimicrobial, has been proposed as a dosing optimization strategy [32]. However, TDM is currently not widely implemented, as the technique is labor-intensive and requires specialized equipment [33]. Furthermore, sample preparation and analysis require a certain turnaround time, impeding instant dosing adjustments at crucial moments. To overcome these issues, we propose to use an AI/ML model that can predict the plasma concentration of piperacillin by using routinely collected healthcare data to provide more real-time antimicrobial plasma concentration information for each patient individually. The used model uses CatBoost, a gradient-boosting decision tree (GBDT) model, and is a variant of the published model in the study of Verhaeghe et al. [34]. Only the top nine features of the a *priori* model were kept to keep the overview in the dashboard. Furthermore, only piperacillin samples were chosen. The model should be integrated in such a way that clinicians can understand the model, interpret the prediction, and explain their decision. Consequently, as the model would be used on an individual patient basis, mainly local XAI methods are ideal for interpreting each prediction.

### 3.2. XAI Methods

There are a multitude of XAI methods. Given the use case, the main focus will be on post hoc, model-agnostic, and local XAI methods for GBDT to explain a single prediction. We focus on both the model-agnostic methods for the generalizability of our research results beyond GBDT, as well as model-aware methods for GBDT specifically, as the back-end ML model is based on the Catboost regressor model. Global XAI methods are not considered as the interest lies in explaining a single prediction. Model-agnostic XAI methods focusing on feature explanations are LIME [10], SHAP [11], and DALEX [23]; all three are suitable for GBDT.

LIME is a model-agnostic feature contribution method that approximates the model locally using a linear model. For every sample, new points are generated around the sample based on the distribution of the training dataset and the model output. These new points are then used to train a linear model that explains the region around that sample [10]. The sample is then put into that local linear model to provide an explanation that shows how every feature in that sample contributes to the given output [10]. SHAP aims to explain the sample by finding the marginal contribution of each feature to all possible combinations of features to explain the output. This means that Shapley values measure how much each feature contributes to the prediction when considered in combination with other features [11]. Compared to SHAP, DALEX measures feature contributions based on the average value of each feature. For every feature, it changes the value to the sample value while keeping other feature values constant at their average value to quantify the contribution [23].

Relevant model-agnostic instance methods for GBDT are Diverse Counterfactual Explanations (DiCE) [24] and Ceteris Paribus (CP) [23]. We also consider leaf influence [22] as a model-aware instance method. DiCE aims to explain why the current prediction is the way it is and not another value by showing new instances with changed feature values that are just enough for the prediction to change substantially, counterfactuals. Counterfactuals are very intuitive for humans to understand, as they work similarly to how we think [24]. CP profiles visualize a function of the prediction based on the change of a single variable. They visualize how a change in a specific feature value impacts the outcome while other features stay constant [23]. Leaf influence determines the most influential training data for GBDTs for the current prediction [22]. However, it is hard to determine whether the calculated influential points are indeed the most influential.

To avoid overwhelming clinicians, we arbitrarily limited the number of applied XAI methods to three to increase the usability of the dashboard. The three selected XAI methods were: SHAP, CP, and leaf influence. SHAP was selected to provide feature contribution explainability instead of LIME and DALEX feature contributions. LIME is not deterministic because of the local sampling of new points and categorizes features into ranges, which impacts the accuracy of the contribution, making it less suited compared to SHAP. DALEX feature contributions and SHAP have the same contribution performances; therefore, as DALEX is less known, we selected SHAP. For the CP curves, we utilized the DALEX library to avoid implementing the method from scratch. The DALEX library offers CP calculation functionality natively using its feature contributions and provides plug-and-play visualizations. CP curves can also somewhat be used as rudimentary local counterfactual explanations. DiCE can provide more expanded counterfactual explanations; however, it does not take real value ranges and feature interactions into account, which makes it less suited for a clinical use case and could confuse physicians. Additionally, leaf influence was included to enable comparison with other cases with the same output to provide more context. Leaf influence is also available by default for CatBoost. Consequently, the selected methods include both model-agnostic and model-specific techniques for local

explanations. Combining feature contribution and instance methods can strengthen the overall interpretation, as these techniques complement each other.

### 3.3. Dashboard Visualization

The constructed dashboard consists of five components and is shown in Figure 1. Each component's content provides specific information on the patient and the prediction, however, due to anonymization requirements, the data has been slightly modified for publication. The first component (Figure 1a) visualizes the current given antimicrobial dose, the predicted antimicrobial concentration, the therapeutic range of the antimicrobial represented as a scale, and where the predicted antimicrobial concentration lies on this scale. The second component shows the feature values of the patient used by the model (Figure 1a). The first and second components are permanently visible. Components 3–5 are, respectively, the SHAP component (Figure 1b), the Ceteris Paribus component (Figure 2a), and the leaf influence component (Figure 2b). Each component visualizes the result of one of the implemented XAI methods. The SHAP component (Figure 1b) shows the impact of each individual feature from component two on the predicted antimicrobial concentration (Prediction $f(x)$), shown as a blue bar. Features that increase the predicted antimicrobial concentration relative to the mean predicted antimicrobial concentration (lowest green bar $E[f(x)]$) are shown in green, while features that do the opposite are shown in red. In this example, one can see that the 8-h creatine clearance forces the prediction to become higher. The Ceteris Paribus component (Figure 2a) explains how the output of the model (y-axis) would change if one single feature were to change (x-axis) while all other features remain the same. The user can select features for which they want such an evaluation, which is then shown as a graph on the right. For example, if the 8-h creatinine clearance had been 150 mL/min, the model output would have been 60 mg/L. The last component (Figure 2b) shows other training samples with similar feature values that influenced the current prediction. The user can activate and hide these three XAI components from view upon preference (they are hidden by default).

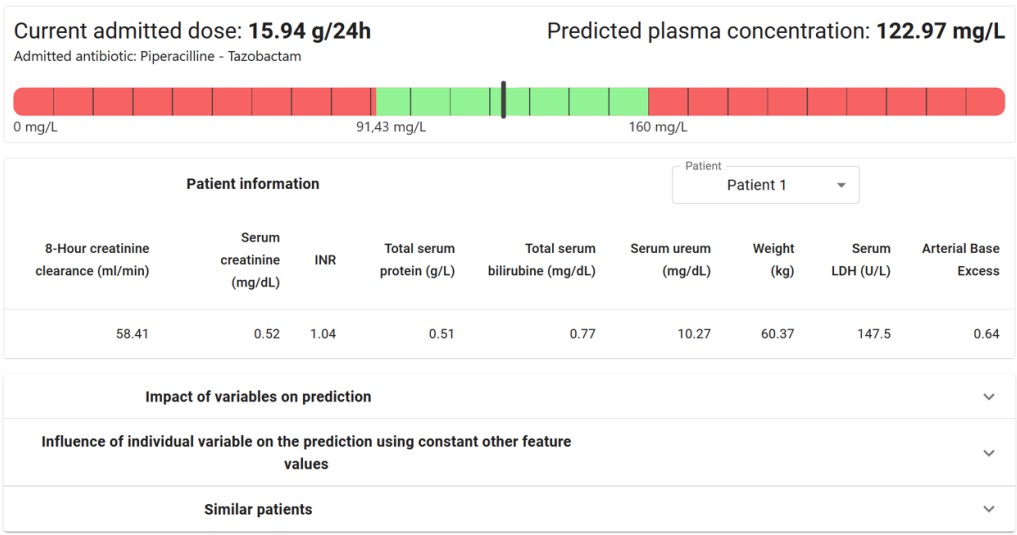

(a)

**Figure 1.** *Cont.*

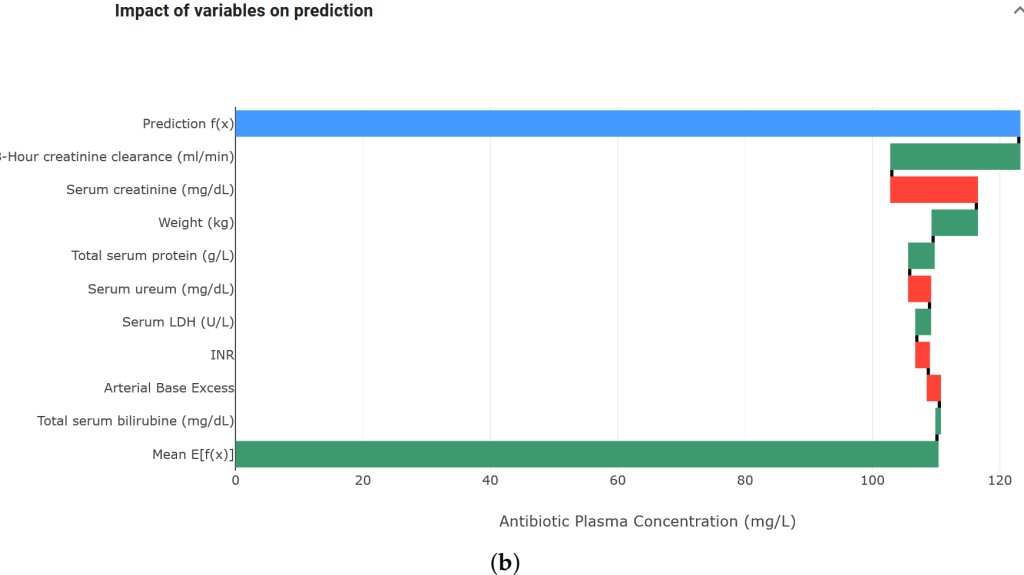

**(b)**

**Figure 1.** The dashboard used in the study with its XAI components. (**a**) The default dashboard visualization of the last given dose, the antimicrobial concentration prediction, the therapeutic range of the patient, and the features of the patient. For privacy reasons for publication, the patient shown here is artificial. (**b**) The SHAP component visualizing the feature values of the considered patient. Red indicates negative shap values, Green indicates positive shap values.

### 3.4. Architecture, Software, and Dashboard Development

The model was available in-house, while the XAI methods were available as Python libraries and integrated into the back end using Flask 2.1.1. Flask is a micro web framework for making simple web applications written in Python [35]. The used Python version was 3.8.10. The used model and XAI method versions were as follows: shap 0.36.0, catboost 1.0.4, and dalex 1.6.0. The final architecture is shown in Figure 3. The front end was developed using React 18.2.0. A React application consists of different components and updates these components individually based on user input, making it an ideal framework for the dashboard for our use case [36]. The back end contains several endpoints that send information as JSON data to the front end. React requests this information on two occasions. First, upon opening the dashboard when all components are first loaded, and second, in the case of user input. The dashboard supports multiple patients so that the user can select and visualize individual patients. The CP profiles can be plotted for every feature of the patient. Therefore, there is the option for the user to select every feature individually to visualize the CP profile of that feature. Each of these actions results in updating the relevant React components. The dashboard was deployed using a single Docker container running Gunicorn 20.1.0 as a static HTTP server. The HTTP server also serves as the React front end for the dashboard.

### 3.5. Participant Recruitment and Dashboard Evaluation Form

Physicians working in the ICU or the clinical microbiology department of Ghent University Hospital, a tertiary teaching hospital in Belgium, were contacted via email to anonymously participate in the evaluation. The recruitment mail provided a password-secured link to the dashboard as well as the evaluation form (both available in the Supplementary Material). Seven clinicians responded and participated in the study.

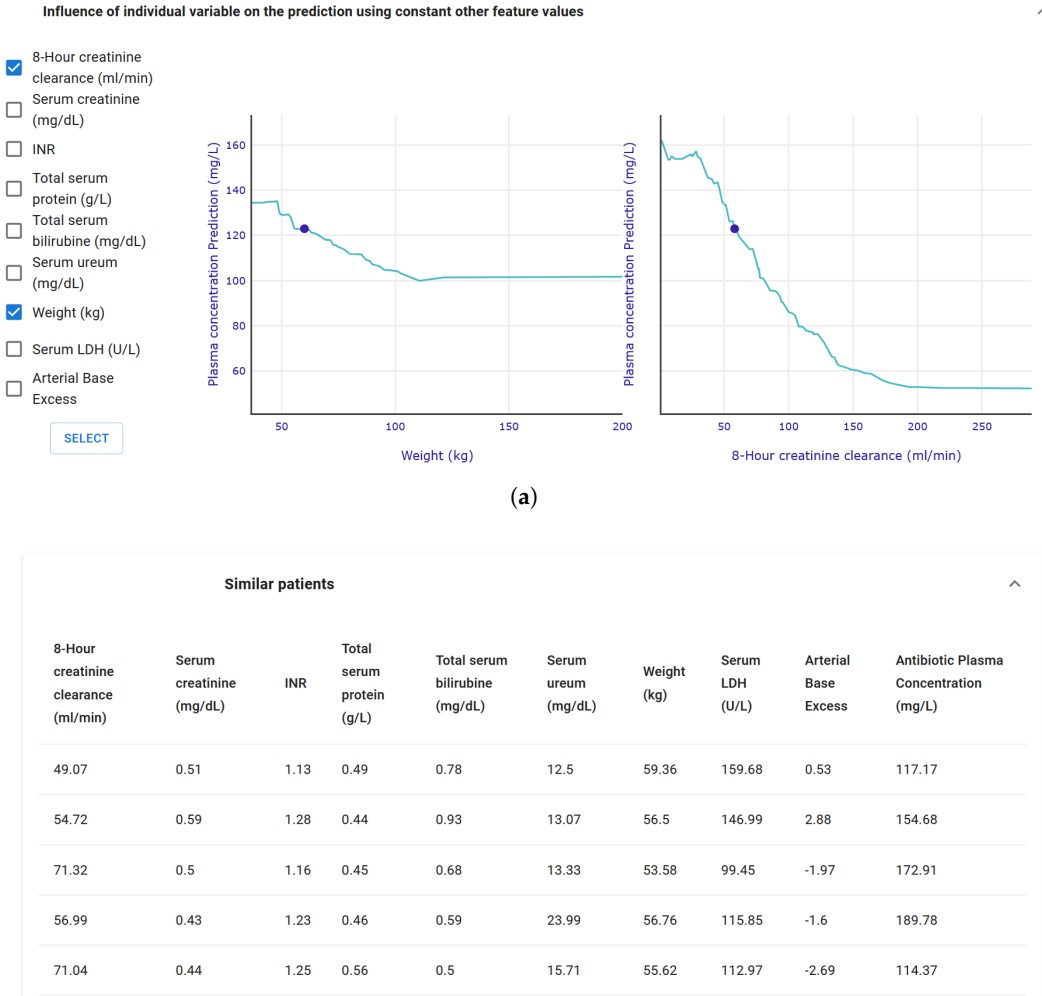

(a)

(b)

**Figure 2.** The dashboard used in the study with its XAI components (continued). (**a**) The Ceteris Paribus component visualizes how the output (y-axis) would change if a feature value (x-axis) were to change while keeping all other features constant. (**b**) The leaf influence component visualizes other patients in the training data with similar feature values that influenced the current prediction. For privacy reasons, the similar patients shown here are artificial for publication purposes.

Five real-world patient cases were presented to the participants. Each of the cases was taken from the same database used for the development and testing of the model (see Supplementary Material; the data has been slightly modified for publication purposes and anonymization requirements). For each case, the features used to produce the predicted antimicrobial concentration, the corresponding predicted antimicrobial concentration, and the corresponding measured antimicrobial concentration were available. The five patient cases were chosen to allow for a mix of concordance and discordance between the measured concentration, the predicted concentration, and whether or not the predicted concentration was in the therapeutic range. The patient cases are summarized in Table 1. For the first patient case, the measured antimicrobial concentration is in accordance with the predicted antimicrobial concentration, which falls in the therapeutic range of the antimicrobial. In the second case, both antimicrobial concentrations are also in accordance and inside the therapeutic range, but close to the lower bound. Cases 3 and 4 represent discordance between the measured and the predicted antimicrobial concentration. While for both cases the measured antimicrobial concentration falls within the therapeutic range, the predicted antimicrobial concentration in case 3 is supratherapeutic, while in case 4 it is subtherapeutic.

In case 5, both concentrations are in accordance and are supratherapeutic. To enable a 'plug-and-play' evaluation of the XAI components, no prior guidance on how to use the dashboard was given. Before completing the evaluation, physicians were asked to give details on the department they worked in and their work experience. For all presented cases, physicians were asked to give the most important conclusions they drew from the dashboard concerning the predicted concentration of the patient. Additionally, they were asked what their treatment plan would be if a single feature were hypothetically changed. This question polls the use of the CP profiles, as these represent hypothetical changes of feature values in the model. Also, for the first and last scenarios, participants were asked to describe their interpretation of each of the five XAI components. These questions represent qualitative questions. After all five scenarios were evaluated, participants had to give a score between one and ten on the contribution of every component to the perceived explainability of the model for a quantitative measure of the interpretability. Additionally, they were asked to score the perceived completeness of the dashboard and the probability of using the dashboard in their clinical decision-making. Responses to the evaluation form were analyzed by two data scientists and a physician familiar with AI and ML applications in healthcare.

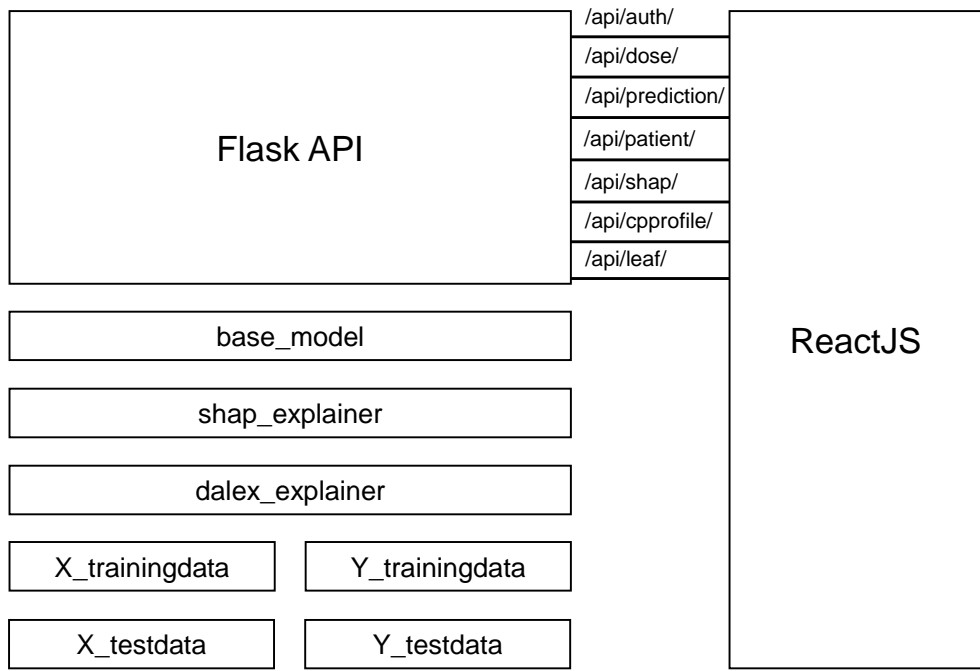

**Figure 3.** System architecture of the dashboard application.

**Table 1.** Overview of presented scenarios.

| Measured Antimicrobial Concentration (mg/L) | Range Measured Concentration | Predicted Antimicrobial Concentration (mg/L) | Range Prediction Concentration |
|---|---|---|---|
| 125.1 | Therapeutic | 122.97 | Therapeutic |
| 90.41 | Lower bound of therapeutic range | 92.03 | Lower bound of therapeutic range |
| 118.95 | Therapeutic | 216.4 | Supratherapeutic |
| 108.8 | Therapeutic | 67.6 | Subtherapeutic |
| 226.4 | Supratherapeutic | 218.7 | Supratherapeutic |

## 4. Results

The averaged scores, as quoted by seven clinicians (six ICU physicians and one clinical microbiologist), for the contribution of the XAI component to explainability are summarized in Table 2.

**Table 2.** Total averaged score per XAI component on contribution to the explainability.

| Component | Score/10 | $\sigma$ |
|---|---|---|
| Predicted concentration | 8.14 | 1.57 |
| Last given dose | 8.14 | 1.57 |
| Toxic–therapeutic margin | 7.57 | 1.51 |
| CP Profile | 7.00 | 1.83 |
| Patient information | 6.29 | 1.70 |
| SHAP | 5.43 | 3.50 |
| Leaf influence | 5.43 | 1.72 |

The three elements with the highest contribution (and hence considered to be the most contributing to the explainability of the GBDT model) were the predicted concentration, the last given dose, and the toxic–therapeutic margin. All three elements are simply output visualizations and thus not XAI methods. Of the XAI methods, the CP profiles were perceived as having the highest contribution to explainability. Derived from the qualitative questions, these CP profiles could correctly be described and interpreted by all physicians. Furthermore, CP profiles were perceived as the most informative, as they provided a way for the physician to estimate the evolution of the concentration given a change in one of the features. SHAP and leaf influence have much lower scores on explainability. Contrary to the CP profiles, several participants indicated that they were unable to fully understand the function of the SHAP representation from the provided visualization, resulting in a large variance in awarded explainability scores. For the leaf influence, participants were able to understand and appreciate that these were similar patients, but could not identify whether these patients were used as a basis for training the model or if they were similar patients for whom the model also provided a prediction. None of the physicians indicated that the prediction in scenarios 3 or 4 could potentially be flawed.

Physicians scored the completeness of explainability of the dashboard with an average score of seven out of ten. Additionally, the physicians scored the likelihood of using the dashboard to support their medical decisions in clinical care at an average rating of 6.29 out of 10. Clarifications and suggestions mainly indicated that additional guidance would be welcomed on how to interpret the XAI visuals, mainly for the SHAP and leaf influence components, as well as specific clinical information for the scenario at hand.

## 5. Discussion

In this work, we set out to evaluate the clinical explainability of XAI methods on end-users for a GBDT model that predicts antimicrobial plasma concentrations of piperacillin. Given the use case and a literature review, we selected three potentially applicable XAI methods: SHAP, CP profiles, and leaf influence. These XAI methods were combined with relevant clinical information in a dashboard and consequently evaluated by clinicians. The evaluation covered five different patient cases, which were presented to six ICU physicians and one physician working in the microbiology department. The physicians gave the complete dashboard a seven out of ten on explainability, which is encouraging for further research. To our knowledge, few of these XAI evaluations with clinicians have been reported in the literature, making this study a valuable step in facilitating AI for the ICU.

Interestingly enough, none of the participants questioned the accuracy and validity of the GBDT model output, as none of the participants identified the predicted concentrations that were discordant from the measured concentrations. Although participants were not encouraged to critically appraise these elements, this observation may indicate a profound challenge for the implementation of AI/ML models. Healthcare workers might not constantly challenge the accuracy and validity of a model's output when used in routine clinical care, as a healthcare worker's main concern is to adequately treat patients in the limited time available. Furthermore, healthcare workers themselves indicate that one of the most prominent expected AI/ML implementation benefits would be that clinical decision support could be provided 24/7, especially at times when human resources are limited or less experienced physicians are the primary caregivers [6]. Therefore, in future work, it is worth exploring whether or not XAI can contribute to the ability of an end-user to identify incorrect predictions.

The top three components contributing the most to the perceived explainability were non-XAI components containing basic context information. Of the XAI methods, only CP profiles were uniformly correctly interpreted and found to be useful. As the CP profiles provide visualizations of how the prediction would have changed if a single feature changed, these visualizations closely approach counterfactual explanations. For humans, these kinds of explanations are the easiest to interpret, as they provide an answer to the "what if" question. The results of our study are in line with the previously mentioned XAI evaluation study by Norrie et al. They evaluated the usefulness of XAI methods, such as SHAP and LIME, with physicians in establishing trust in an ML model that predicts a sepsis diagnosis. Our study confirms these results for SHAP and places the Leaf influence component on the same level, while the CP profiles have a much higher perceived explainability. Norrie et al. reported neutral to negative scores for SHAP and LIME methods when it came to user trust, explanation satisfaction, and human–machine task performance. These scores were given in their study, even though participating physicians received information regarding these techniques at the beginning of the study [16]. To our knowledge, no study evaluates the perceived explainability of CP profiles with physicians.

The SHAP and Leaf influence results indicate that using popular XAI methods 'as is' might be insufficient, even with a short, readable explanation, to adequately confer the explainability physicians want and expect. In other user groups, other less common XAI methods are more suited for use 'as is'. Kaur and colleagues found that explanations provided by generative additive models (GAMs) are easier to understand intuitively than explanations provided by SHAP; however, their study population was limited to data scientists [37]. They mainly attributed their findings to the design of the XAI method output instead of the participant's interpretation. Notably, few of the data scientists could accurately describe the output visualizations of both GAM and SHAP. If data scientists themselves cannot accurately interpret XAI methods when given 'as is', additional clarification techniques might be necessary to realize utility for lay end-users.

Natural language explanations are another way of providing explanations, which was not researched in this study. A study where additional explanation of the XAI methods was provided using natural language has shown to improve user decision-making by 44% and might therefore improve perceived explainability of the XAI methods [38]. Several interactive XAI systems that leverage natural language are currently being researched [39,40]. Additionally, combining natural language explanation with preattentive processing has also been shown to improve the end-user's understanding of a model's behavior [41].

Our study also has some limitations. Only a limited number of participants from the same hospital took part, increasing the chance of selection bias. Additionally, a single composition of the dashboard was tested, which might have limited the potential of the

dashboard. Finally, since the setup of the study was to investigate the perceived explainability, no conclusions can be made about the potential trust in the model. To evaluate the XAI dashboard on a broader range of criteria beyond pure interpretability, future work should supplement the current qualitative questionnaire with quantitative evaluation indicators; examples include time to decision, system usability, and user interaction metrics.

Future work could be directed toward improvement of the research methodology, optimization of the dashboard, and broadening of the scope of the research questions. Expanding the testing population to physicians working in different specialties, as well as to a multicentric setting, will increase research validity and potentially uncover additional insights. Furthermore, since CP profiles were perceived as the most clinically useful, this XAI method is worth exploring further to fully understand its potential in a clinical setting. The effect of changes to the dashboard, such as different components, layout, and the addition of relevant clinical information, on the perceived explainability will also further concretize the requirements and the needs of clinicians. To evaluate the XAI dashboard on a broader range of criteria beyond pure interpretability, future work should supplement the current qualitative questionnaire with quantitative metrics; examples include time to decision, system usability, and user interaction metrics. Finally, the scope of XAI research should be broadened. Not only should the perceived explainability by the end-users be investigated, but also the possibility of end-users identifying incorrect predictions through XAI and whether or not XAI methods elicit trust in end-users to start using the AI/ML model. In general, many more studies are required to fully comprehend the requirements of AI for use in an ICU setting in terms of trust, explainability, and perceived usefulness, for which this study can serve as a stepping stone. Hence, this study could be seen as a reference for future XAI research and clinical implementation.

## 6. Conclusions

In conclusion, a first attempt was taken to formally evaluate the usability of XAI methods to satisfy the explainability requirement posed by end-users to start using AI/ML models in high-stakes environments such as the ICU. With an average score of seven out of ten for explainability, the responses from seven practicing physicians underscore the potential impact of an XAI dashboard on clinical decision-making, with participants specifically highlighting the CP profiles as the most useful XAI technique. Although small, this work can serve as a starting point for the exploration of XAI applications to foster and ultimately reach AI/ML implementation in clinical ICU practice.

**Supplementary Materials:** The following supporting information can be downloaded at https://www.mdpi.com/article/10.3390/app152010933/s1: Document S1: Case Report Form; Document S2: Dashboard supplement of all five cases.

**Author Contributions:** T.D.C. and J.V. share the first authorship; conceptualization, T.D.C. and J.V.; software, J.V.; validation, T.D.C., J.V., F.O. and S.V.H.; formal analysis, T.D.C. and J.V.; investigation, T.D.C. and J.V.; resources, J.J.D.W., F.O. and S.V.H.; data curation, T.D.C. and J.V.; writing—original draft preparation, T.D.C. and J.V.; writing— review and editing, T.D.C., J.V., F.O., J.J.D.W. and S.V.H.; visualization, J.V.; supervision, F.O., J.J.D.W. and S.V.H.; project administration, F.O., J.J.D.W. and S.V.H.; funding acquisition, J.V., F.O., J.J.D.W. and S.V.H. All authors have read and agreed to the published version of the manuscript.

**Funding:** This research was funded by the FWO Junior Research project HEROI2C, which investigates hybrid machine learning for improved infection management in critically ill patients (Ref. G085920N), and by the Flemish Government under the "Onderzoeksprogramma Artificiële Intelligentie (AI) Vlaanderen" program. Jarne Verhaeghe is funded by the Research Foundation Flanders (FWO, Ref. 1S59522N). Jan De Waele is a senior clinical investigator funded by the Research Foundation Flanders (FWO, Ref. 1881020N).

**Institutional Review Board Statement:** The study was conducted in accordance with the Declaration of Helsinki and approved by the Ethics Committee of Ghent University Hospital (protocol code: BC-03197; date of approval: 19 May 2022).

**Informed Consent Statement:** Written informed consent was obtained from all respondents involved in the study to publish the aggregated results across the respondents. Written consent was deemed sufficient by the Ethics Committee, as no individual details are published, and only consent was required to process their information.

**Data Availability Statement:** The data presented in this study are available from the corresponding author upon request due to privacy and data sharing agreement limitations.

**Conflicts of Interest:** The authors declare no conflicts of interest.

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
