# Peer review of "Towards Trustful Machine Learning for Antimicrobial Therapy Using an Explainable Artificial Intelligence Dashboard"

_applsci, doi:10.3390/app152010933_

Round 1
Reviewer 1 Report
Comments and Suggestions for Authors
This research topic is at the forefront and closely aligns with clinical needs. It focuses on the interpretability of AI/ML models in the high-risk environment of the ICU, and conducts a study on the optimization of piperacillin dosage. It has high practical value. Multiple XAI methods (SHAP, CP, Leaf Influence) are integrated into the dashboard, and a questionnaire assessment is conducted by seven clinical doctors on five real patient cases, demonstrating the user-centered design concept. The results show that the CP curve is the most recognized in terms of interpretability, and the average score of the overall dashboard interpretability is 7/10. This provides preliminary empirical support for XAI in ICU clinical applications, and also reflects the differences in doctors' understanding of XAI visualization. The research is innovative in the practical application of local XAI methods combined with feature contributions and instance methods, and provides a reference for the optimization design of XAI dashboards in the future.
However, this study has obvious limitations. The sample size is small (five patient cases, seven doctors), and it only focuses on a single antibacterial drug, which affects the generalizability and representativeness of the results; the interpretability assessment mainly relies on subjective questionnaires, lacking quantitative indicators; and the actual impact of XAI dashboards on clinical decision-making quality or patient outcomes has not been evaluated. It is recommended to expand the sample size, diversify patient scenarios, introduce quantitative evaluation indicators, and verify the value of XAI for decision-making and treatment outcomes through simulation or real clinical trials. Overall, this study can be regarded as exploratory work on XAI in ICU clinical applications, and is suitable as a reference for future large-scale research and clinical implementation.
Author Response
We sincerely thank the reviewer for their time and constructive feedback. We are encouraged that the reviewer recognized the study's innovative nature, its alignment with clinical needs, the value of integrating multiple XAI methods, and the user-centered design concept.
The reviewer correctly identifies the study's limitations, including the sample size, single-drug focus, and reliance on qualitative assessment. We agree with this assessment and appreciate the reviewer's insight that "this study can be regarded as exploratory work on XAI in ICU clinical applications." This exploratory scope was our precise intention, hence the current study design.
While quantitative evaluation indicators are essential for a full impact analysis, the primary goal of this initial study was to first assess the core interpretability of the XAI methods from the clinicians' perspective. These findings aim to provide, as the reviewer notes, "a reference for the optimization design of XAI dashboards in the future."
To better contextualize our contribution and address the reviewer's points, we have added two sentences to the Discussion section. First, we have expanded on future work by stating: "To evaluate the XAI dashboard on a broader range of criteria beyond pure interpretability, future work should supplement the current qualitative questionnaire with quantitative evaluation indicators, examples include time to decision, system usability, and user interaction metrics." Second, to properly frame the study's contribution, we have concluded the section with the sentence: "Hence, this study could be seen as a reference for future XAI research and clinical implementation."
Reviewer 2 Report
Comments and Suggestions for Authors
Dear Authors,
Thank you for a very well presented and well executed study that will provide great value to health care and safe and quality care with emerging technology.
The opening sections 1 and 2 provide well balanced and explained information for the reader followed by an equally comprehensive yet succinct methodology.
Please consider these points to potentially strengthen the presentation and clarity of the key information provided:
- Section 3.5 doesn’t state how many physicians were recruited? The first line of the results (4) confirm this however this should be stated within methods
- Line 253-258 explains the data was quantitative in being ratings summarised in Table 2. However additional (line 254) quantitative measures of ‘completeness’ of the dashboard, ‘probability of using the dashboard, and ‘suggestions’ which would be qualitative, do not appear to be represented in the results?
- Line 274-278 – if the leaf influence rating was the data collected, how were the researchers able to observe the comments made here? The above point may relate to this?
- Line 279-284 – where is this data?
Author Response
We thank the reviewer for their feedback, time, and constructive comments, which have helped us improve the clarity and presentation of our work. Please find our point-by-point response below.
Reviewer's Comment 1: Section 3.5 doesn’t state how many physicians were recruited? The first line of the results (4) confirm this however this should be stated within methods.
Our Response: We have added the sentence, "Seven clinicians responded and participated in the study." at the beginning of Section 3.5, as this should indeed be stated within the methods.
Reviewer's Comment 2 & 4: Line 253-258 explains the data was quantitative in being ratings summarised in Table 2. However additional (line 254) quantitative measures of ‘completeness’ of the dashboard, ‘probability of using the dashboard, and ‘suggestions’ which would be qualitative, do not appear to be represented in the results? ... Line 279-284 – where is this data?
Our Response: We thank the reviewer for highlighting this ambiguity.
-
The quantitative results for the ‘completeness’ and ‘probability of using the dashboard’ measures are presented in the final paragraph of the Results section. Specifically: "Physicians rated the completeness of explainability of the dashboard with an average score of 7 out of 10... When asked about the likelihood of using the dashboard... physicians gave an average rating of 6.29 out of 10." To improve clarity, we have slightly rewritten this section to explicitly mention that these are the scores: "Physicians scored the completeness of explainability of the dashboard with an average score of 7 out of 10. Additionally, the physicians scored the likelihood of using the dashboard to support their medical decisions in clinical care at an average rating of 6.29 out of 10. Clarifications and suggestions mainly indicated that additional guidance would be welcomed on how to interpret the XAI visuals, mainly for the SHAP and leaf influence components, as well as specific clinical information for the scenario at hand."
-
Regarding the qualitative data on ‘suggestions’, we have consolidated the common theme in the response mentioned above, and more explicitly stated that these were the suggestions and the clarifications.
Reviewer's Comment 3: Line 274-278 – if the leaf influence rating was the data collected, how were the researchers able to observe the comments made here?
Our Response: Our questionnaire, provided in the supplemental material, was designed to capture mixed-methods data. For each XAI component, participants were first given open-ended questions to elicit qualitative feedback and reasoning, followed by providing a quantitative rating of the XAI interpretability. The comments presented in lines 274-284 are a synthesis of the qualitative data gathered from these open-ended questions. We have added "Derived from the qualitative questions," to this result section to indeed avoid the confusion. Thank you for pointing this out.